# Newman and Wittgenstein on the Will to Believe: Quasi-Fideism and the Ground of Religious Certainty

Modesto Gómez-Alonso 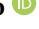

Department of Metaphysics and Current Trends in Philosophy, Ethics and Political Philosophy, University of Seville, 41004 Sevilla, Spain; mgomez26@us.es

**Abstract:** In this article, I argue that Newman's emphasis on a gestaltic model of reasoning and the role played by the imagination in informal reasoning is a fruitful starting point for an exploration of convergence between the *Grammar of Assent* and Wittgenstein's *On Certainty*. I claim that Wittgenstein, like Newman, challenges both the claim that justification must be neutral and the prejudice according to which any form of persuasion that is not demonstrative is irrational or arational. Arguments are mainly focused on the picture of Newman's epistemology provided lately by Duncan Pritchard. I argue that Pritchard misrepresents Newman's conception of the illative sense so as to ascribe to him the thesis that religious belief is evidentially grounded in a broad sense of evidence. This creates a false dichotomy between the arational view of religious principles and the account of religious certainties as epistemically grounded. I suggest that Newman's reference to both living persuasion and the role played by the will in religious conviction is part of his attempt to expose this false dichotomy.

**Keywords:** Newman; Pritchard; Wittgenstein; assent; ethics of belief; hinge epistemology; illative sense; inference; paradox; religious conviction

## 1. Introduction

One way of approaching Wittgenstein's thought in *On Certainty* is suggested by the discussions initiated by Newman's anti-evidentialism and his rejection of the Lockean thesis that assent, in order to be rational, must be proportional to evidence. However, my comparison between these two philosophers should be prefaced by some words of caution.

First, Newman is commonly placed alongside Descartes and Kierkegaard as a strong volitionalist who proposed a free, unconstrained act of the will as a way to bridge the gap between conditional inference and unconditional belief (Pojman 1978, p. 2). I will claim that this view throws a false light on Newman's thought as a whole.

Secondly, reference to Wittgenstein's last notes is particularly problematic. This is due, at least, to two factors: (a) the first is the fact that, beyond the claim that hinges are not discovered as metaphysical matters of fact, consensus is lacking, both regarding the epistemic standing of hinges (if any) and whether genuine hinge certainties should be distinguished from other kinds of everyday certainties; and (b) the second is the difficulty of dissociating Wittgenstein's conception of religious belief (viewed as a particular application of hinge epistemology) from fideist or quasi-fideist positions, according to which faith is in no way anchored in the living, first-personal process that precedes it. Since I do not think that Wittgenstein's position excludes the possibility of a sort of retrospective, engaged justification of religious belief (one which is not based on 'neutral' evidence), I will not provide support for the absolute incommesurability of religious frameworks (and

thus, for a picture of religious commitment as invulnerable to criticism) that fideism and quasi-fideism entail.

Finally, though the question of religious belief and the main topics of hinge epistemology are not entirely separable, I will pay more attention in this paper to the latter. This is because, among other reasons, Newman's method indicates that an understanding of religious belief is dependent on an understanding of belief and certainty in general.

However, the main stimulus for this article comes from the picture of Newman's epistemology provided lately by Duncan Pritchard, one that aims to emphasize the divergences between Wittgenstein's thought and Newman's alleged proto-virtue epistemology (Pritchard forthcoming). It is not only the case that Pritchard has come to reject his early view that Wittgenstein's dealing with radical scepticism is a way of working through the implications of Newman's ideas (Pritchard 2015, p. 425). It is also that Pritchard, in his new approach, misrepresents Newman's conception of the illative sense so as to ascribe to him the thesis that belief is evidentially grounded *all the way down*, even if, in this context, evidence is understood in a broad sense that includes, against the Lockean austere conception of evidence, *extraordinary* or peculiar kinds of evidence of a personal and subjective nature.

Pritchard's shift from a quasi-fideist reading of Newman, with emphasis on *discontinuity*, to a new approach that targets Newman's emphasis on *continuity* is, in my view, highly suggestive. What it suggests is an implicit, background model of reasoning, aggregative, factorising, and mechanical, that is shared by direct volitionalists and evidentialists (whatever their kind) alike and that keeps us hovering between the 'rational aspect' and the 'arational aspect' of a 'rational–arational' figure.

These two aspects seem to exhaust our ways of establishing the relationship between assent and inference since they are grounded in a picture of certainty as though it is the result of a logical relation between premises and conclusions which, if it falls short, must be supplemented with an extra pull. It is as if the logical gulf between conditional premises and unconditional assent that Newman describes would leave only two options for filling the gap: either assent is resolvable into its reasons, so as to come by degrees, or unconditional assent is the result of an arbitrary choice (rephrased in Kierkegaard's idiom: "of a *leap*"). In terms of *qualitative transition*, it is as if one were either bound to conserve the transition at the cost of making it a *gradual* and *quantitative* one, or to cancel the fluid nature of the transition by means of a *fiat* external to the process that leads to conviction, where a 'fiat' stands also for any arbitrary mechanism, arational or visceral, for generating commitment. However, the effectiveness of Newman's informal proofs is crucially dependent on their being not regarded as proofs in the sense of traditional proofs.

It is my view that Newman attempted to *dispel the above illusion* by attacking its source and that he aimed at exposing a false dichotomy and to steer a way between the rational and the unconditional. Newman describes a particular mode of adherence and belief, one that allows critizability but which is also unconditional, in which belief does not mean an intellectual assent to isolated, knowledge-apt propositions. This means that belief is not a propositional attitude that is a constituent part of knowledge. This is precisely the function of the illative sense. Curiously enough, it is Newman's insight that certainty is not exhausted by a logical paradigm that might allow us to come to forget, as Wittgenstein recommended, "this transcendent certainty" (Wittgenstein 2004,[1] §[2] 47)—a 'transcendent certainty' to which the arational, the inefabilist, and the transcendentalist accounts of hinges are equally bound.

For Newman, 'certainty' is, as it happens with the 'illative sense', a "grand name for a common thing" (Newman 2015, p. 436).[3] Newman's certainties are humdrum certainties that are grounded in our life. They are not 'transcendent' because they are not external to the process of informal reasoning that, through the middle of ordinary life, leads to them.

Neither are they fully immanent to the process, on pain of viewing them as grounded in so-called 'evidence', and thus, as the compulsive affirmations of the rational automaton.

If the following proposal holds water, then similitudes between Newman and Wittgenstein may be more expansive and deeper than a cursory glance on their thought might suggest.

## 2. Newman's Paradox

Let us begin by noticing that since assent is, for Newman, the assurance we naturally and directly feel, I will use 'assent', 'belief', and 'certitude' as synonyms. It is then important to distinguish Newman's technical understanding of 'certitude' as a complex, reflective assent (Newman 2008,[4] pp. 157–80), namely, as a deliberate commitment to an already experienced certitude, from the first-order, simple certainty we feel when believing.[5] It is the nature of the latter that I aim to discern in the present text.

It is common knowledge that Newman stressed the absolute, unreserved character of certainty and that he also claimed that certitude is a natural state, even a *normal* state, of the mind. Finally, he was emphatic on the point that certainty is a *sui generis* state—neither *logically continuous with* nor *reducible to* reasoning and neither susceptible to demonstration nor arbitrary and empty.

Newman's claim that assent is unconditional[6] and certainty is natural has been usually understood as leading straight to a paradox. The paradox runs as follows—if Newman is right, then, regarding belief for which evidence is less than conclusive, only two rational alternatives are open for us: either is withheld (scepticism), or else there is an assent that rises higher than its source, and which is hence irrational. Thus, if one is to adopt a formal and logical model of reasoning, then one must always choose between helpless agnosticism and irrational commitment. There would be, however, a third alternative, that would make it possible to escape from Newman's paradox. One may take a *pragmatic* approach to the question, and thus, one might claim that when evidence, though not conclusive, is favourable for p, we can give a tentative, qualified assent to that proposition, making thus provision for further doubt. On this view, hesitant assent (which, for Newman, is not assent at all) would be sufficient to conduct our epistemic practices.[7]

It might be plausible to contend that the main aim of hinge epistemology is to provide an epistemic (and not a merely pragmatic) response to the above paradox.[8] To this end, hinge epistemologists remind us that the epistemic status of hinge certainties is a function of centrality in our web of beliefs and that some principles constitutive of rationality, though groundless, are framework principles applicable to all beliefs, in any context. Hinges may well be relieved of the need of warrant, but, in this view, they are not relieved of the need of some sort of a priori justification.

It is clear that if Newman's exploration of reasoning focused the above paradox, then his investigation would overlap neatly with hinge epistemology. However, it is my view that taking this epistemological shortcut would be a false step that would lead us to misrepresentation. It is not only the case that Newman's paradox, far from being phrased from the standpoint of the mandates of reason and epistemic rationality, i.e., far from being an idealized form of paradox, is a humble description of natural reasoning. Rather, the paradox faced by hinge epistemologists urges us to respond to it without dispelling its illusion.

My point is that while Newman's interest lies in describing how our minds actually operate in the very process of reaching certitude, hinge epistemologists are focused on how certitude might be speculatively justified after it is experienced, meaning that hinge epistemology is a second-order project which is mainly concerned with offering a defence of the rationality of belief in general in the face of the challenge of radical skepticism. However, for Newman, judgments of certainty are tied to ways of living. For him, we must

attempt to appropriate the paradoxical tension without trying to resolve or to eliminate it by abstract means. I have the impression that most hinge epistemologists remain hostage to a *residuum of doubt*, when genuine doubt, doubt which operates in our informal reasoning, is effectively discarded in the process of reasoning. Thus, Newman is inviting us to go back to the rough soil of the paradox, and hence to abandon (at least at this level of scrutiny) a "further doubt behind" genuine doubt (OC, §19) and the etiolated sort of normative justification that it elicits. This is why Newman's exploration of *informal* reasoning focuses its *sui generis* paradox, one which, I repeat, should not be conflated with its epistemological surrogate, however much they may be ultimately related.

What, then, is Newman's paradox? It is, in his own words, "the inconsistency which is involved in holding that an unconditional acceptance of a proposition can be the result of its conditional acceptance" (GA, p. 135). The paradox thus lies in that excess of belief over proof that Newman illustrates with a battery of everyday certainties which, contrary to the Lockean principle of proportionality, are supported by an array of reasons that *do not touch them logically* (GA, p. 254). The similitudes between Newman's and Wittgenstein's examples have been noticed, and for good reason. It may also be noted that Wittgenstein was fully aware of this discontinuity, and that, in the case of religious belief, he even *extended* it, taking notice of the disproportion between the 'enormous things' to which believers give their assent and the 'flimsy' evidence on which it is, *in a certain sense*, based (Wittgenstein 2007,[9] pp. 57–58).

There is, however, something odd in how Newman deals with the paradox. One might expect to find a *direct* answer to it in the *Grammar of Assent*. All our efforts are, however, to no avail. After presenting the paradox, Newman, instead of providing a clear-cut response to it, plunges himself into a long discussion of informal and natural apprehension that culminates in a chapter about the illative sense. This suggests that *Newman's discussion of informal reasoning and the illative sense* is itself somehow the answer to the paradox—a suggestion that, *taken in one way,* is correct. The difficulty now lies in identifying the wrong way of taking this answer, and in arguing why it is wrong.

The concrete action of our minds involves the use of the illative sense. Newman uses this term to refer to the power of reasoning in general, as it operates informally and naturally. Though it does not provide a logical proof, the illative sense can lead, as a result of converging probabilities, to genuine certainty.

The temptation is now to reconstruct Newman's conception of the illative sense as though it were a cognitive virtue that, when reliable, would lead the agent to knowledge (Pritchard forthcoming). This interpretation, as Pritchard proposes it, turns around three main claims:

(a) Newman's position differs from traditional evidentialism in a more expansive conception of evidence. Since *extended evidence* includes personal evidence that is not publicly accessible, this reading can easily accommodate Newman's anti-evidentialism (of a sort), as well as his view that the reasoner, while rational, might be unable to communicate his reasons directly and to submit them for public assessment. This means that the agent might be unable to give the reasons that he has. This view would thus emphasize the parallels with Moore, who puts forth the same claim that there are many things we know yet for which we cannot give our reasons (Moore 1959, p. 149).

(b) It is true that Newman's expansive conception of evidence runs against what, for most epistemologists, counts as evidence. However, Newman's evidence is still genuine evidence. This means that, for Pritchard, Newman contends that our quotidian certainties, if properly formed, are rational and knowledgeable. They are, contrary to Wittgensteinian certainties, in the market of knowledge.

(c) The illative sense is a sort of holistic sensibility to relevant evidence, a skill that, while it cannot be codified, is, like Aristotelian prudence in the practical domain, fully operative in our intellectual endeavours.[10] Again, in this view, the illative sense is the *intellectual* power (within particular domains of performance) to appreciate the *force* of available reasons and how they relate, on their own, to conclusions. We can, indeed, cultivate this power, but the force of the argument still lies in the argument itself, and not in how the reasoner relates *actively* to it. In this reading, the personal dimension of the process leading to certainty seems to refer, at most, to the personal experiences that the process now includes, and to the personal cultivation of our sense of evidential antecedents and of how knowledge is necessarily elicited by the intrinsic force of argument.

It is true that Newman introduces the informal method of reasoning by calling attention to the fact that it includes "probabilities too fine to avail separately, too subtle and circuitous to be convertible into syllogisms" (GA, p. 256). So, it makes sense to ascribe to him an expansive conception of reasoning which highlights its implicitness. It is my view, however, that Pritchard is blind to the kind of thing that Newman was wanting to deny, and that, unawares, his version of Newman is just a portrayal of the intellectual opponent Newman saw himself facing.

First, the reasons that lead to certainty cannot be meaningfully described as *evidence*, neither publicly available nor implicit and personal. Pritchard's picture suggests that if we were able to occupy the exact epistemic position of the believer, and thus, to survey the whole array of his experiences, public and personal, and to appreciate them with the same degree of illative competence, we would be *rationally forced*, like the believer is, to adopt his belief. In this view of Newman, belief is the *compulsory effect* of the reasoning process, a passive event that just comes to happen to us in the light of, and by the force of, available evidence.

This not only clashes with Newman's striking affirmation that "certitude is *not a passive impression made upon the mind from without*, by argumentative compulsion" (GA, p. 271), but also with his portrait of the role played by the *personal* in reasoning, where the *content* of our reasons cannot be abstracted from the personal and the attitudinal, and one cannot be reduced to the other. This is one of the aspects of what I meant above by 'appropriating the paradoxical tension'. This is also why Newman is so adamant in his claims that *it is the person who reasons* (GA, p. 66), and that *we do not become certain without or against our will* (GA, p. 189). Again, the act of assent is not, for Newman, impersonally compelled by arguments. Regretfully, Pritchard seems to conceive Newman's reasoner as a mere spectator of his passive intellectual processes.

These last remarks lead us directly to my second consideration regarding the illative sense.

It is far from the truth that the illative sense, however much it may be grounded in virtues of character, is a virtue of the barren intellect. On the contrary, for Newman, it is a power of the *trained imagination* (GA, p. 250),[11] the potential for the active recognition of *patterns* which are not explicitly present, as its direction, in neutral evidence. The crux of the matter is that, for Newman, the conclusions of informal reasoning are not compelled and not arbitrary, but rather *interpreted* (GA, p. 131). This means that the true force of a non-demonstrative argument does not lie in how the premises relate to the conclusion, but rather in how the reasoner personally relates to those premises by means of what may be called an 'imaginative elucidation'. It is precisely where logic, evidence, and knowledge—all of them—fail that reasoners become personal and appeal to "their own illative sense" (GA, p. 288). Thus, the illative sense may well have *persuasive effectiveness*, but it will never have cognitive effectiveness.

This changes the paradigm, since what Newman is describing is a process that, though it is anchored in reason, culminates in a sort of imaginatively driven, synthetic *gestalt shift* that gives to itself an intellectual foundation only *retrospectively*. It is, therefore, as if, although reasons are registered during the process that leads to a *change of posture*, they acquire their full *significance* only after the shift. They are not cumulative reasons; they are reasons insofar as they are internally related to an already discerned pattern. This illuminates Newman's admissions that "what for one intellect is a proof is not so to another" (GA, p. 233), and that, in regard to certitude, which is *the understanding of one's own experience*, we can only speak for ourselves.

It should not seem strange that Newman makes abundant use of gestaltic comparisons in which it is the imagination that makes things concrete and determinate, such as the way in which a portrait fleshes out a mere sketch, or the accommodation of the eye to "catch and lose again" (GA, p. 249) aspects hidden in a landscape, or the expanding polygon with its asymptotic approach to the imaginative limit of a circle (GA, p. 253). Continuity and discontinuity are equally emphasized in those examples, whose function is to illustrate how certitude does not come under the reasoning faculty, but under the jurisdiction of the imagination.

In short, Newman's response to the paradox is a *paradoxical response*. This is not because Newman proposes a paradoxical way of escaping the paradox (the leap into the arational), but rather because the paradox is displaced from a free act of the will to the recognition of the paradoxical, and thus, willing nature of informal apprehension. Against the backdrop of the linear, cumulative, and logical model of reasoning, 'seeing as' and 'noticing aspects' are paradoxical phenomena since they are non-standard, imaginative ways of apprehending truths. There is no sense in which certainties acquired in this way can be cognitive *achievements* and thus be in the epistemological market of knowledge. For them, this market is normatively closed.

## 3. Newman's Gestaltic Turn

Since Newman is anxious to point out that reasoning is not impersonal or compelled, we do not become certain without or against our will, as it was noted earlier.

This reference to the will parallels the 'willing nature' that William James highlights in his famous essay on the *will to belief*—the passional nature, the "factors of belief" which complement our intellectual nature (James 1956, p. 11). This commitment to passion, shared by Kierkegaard, Newman, James, and Wittgenstein, points to the role of imagination in informal reasoning. However, it is also important to notice that while James' main objective is to argue that we are *within our epistemic rights* in believing something when the options are living, forced, and momentous, and when the question cannot be decided by *intellectual grounds* (James 1956, p. 29), Newman is far from understanding the will, like James does, as a gap-filler which is rationally permitted under certain conditions. Indeed, he is far from attempting to argue for our *right* to believe.

Certainty may well be person-variable, but for Newman it is also absolute and complete, irrevocable, and unconditional. Thus, from Newman's position, James' program misfires in two distinct ways: it misdescribes the process leading to certainty by attaching an independent act of will to reasoning—as if, because reasoning precedes will, they were two independent factors—and it conflates the first-order explanation of how we become certain with the project of providing a rational, reflective certificate to our certainties (GA, p. 162). One common mistake in approaching Newman results from insufficient appreciation of the distinction that he made between those two projects.

The problem is now how to reconcile the two claims on which the whole of Newman's position rests: the claim that certainty is a *sui generis* state, which is not resolvable into its

antecedents, and the claim that certainty is also prefigured in reasons which are intimations of belief—since belief does not come by means of a *fiat* after deliberation.

What, then, is the role played by the will in reasoning when the transition from inference to belief is non-volitional? The crux of the matter is that a model should be provided that is able to make the *dialectical tension* between continuity and discontinuity intelligible. Additionally, this model should make sense of the description of the process, leading to assent being "under the jurisdiction of the will", while acts of the will are ruled out of the picture. How can it be that believing that *p* is not a free choice at all, and yet it is it not a forced one either?

The solution is expressed, as was already hinted, in terms of a 'gestalt shift'.

Let us focus on simple gestaltic figures such as the 'duck–rabbit' figure. The first thing we can note is that, though noticing, say, the 'duck aspect' of the figure, may take a preliminary effort of concentrated attention, and maybe even some kind of linguistic guidance, the seeing comes, as it were, *of itself*. The attending is a free act, yet the recognition we have is not achievable by a decision. This not only means that coming to notice the aspect is not the result of direct volition (as if we could choose to see the aspect, and not only, as it happens, to *try* to see the aspect), but also, that this is an all-or-nothing phenomenon, a *decisive* transition with the same marks of absoluteness and unconditionality, lack of degrees, and conclusiveness with which Newman describes certainty. Far from coming in degrees, the recognition of the 'duck aspect' of the figure is a sudden, holistic, and complete "taking hold" (OC, § 511). This point is further reinforced by the fact that it is not the object that changes, as if new lines and shadows were added to it, but rather what it changes is our way of seeing it. Discontinuity, then, is a necessary property of gestaltic changes.

This does not entail, however, that there is not an *intrinsic* continuity between the object and the noticing of its aspects, so that the shift is not integrally related, as it is, to what goes before. For one, the shift depends upon the material properties of the painting, to the point that, as Wittgenstein highlights, in the cases of musical or poetical appraisal, the change of a simple note, or of a word, or of intonation, might result, for good or for bad, in a different appreciation of the whole work (LC, pp. 37–38). Those are quotidian examples of how 'enormous things' could hang legitimately on 'flimsy' data.

Additionally, and though Newman would say that, after reaching the point of the shift, there would be no 'more or less' in seeing it as a duck, one could also follow with the tips of one's finger the 'duck lines' of the object, which have been registered during the process. The point is that, although the 'gestalt shift' is not quantitative, it is nevertheless a function of what precedes it. As was previously said, one is always able to provide a retrospective, post hoc kind of justification of what one sees, even though it is also true that one could "never have arrived to belief *[or, by the same token, to see the 'duck-aspect' of the figure]* by way of such proofs" (Wittgenstein 2006,[12] p. 97).

Finally, let us mention, even if in passing, that 'aspect blindness' is a common phenomenon—one that is plausibly related to the imagination (or its absence). The crucial point here is that no one can be compelled to see an aspect, just like no one can be compelled to *feel certain* of something, even when confronted with arguments that lead others to conviction. It is in this sense that 'noticing an aspect', like certainty, is *free*: it cannot happen *without* or *against* the will; it is incompatible with compulsion; it is *irreducibly first-personal*. This is why Newman—because he is illuminating informal reasoning and certainty from a 'gestalt shift' model and because he is seeing the illative sense, as the imaginative power of forming and recognizing *patterns*, as the synthetic activity of giving order to a heap of disconnected facts—contends that he is dealing, not with speculative certainty, with practical certainty, or demonstrative certainty, but with *moral* and *interpretative* certainty (GA, p. 252).

H.H. Price argued that the distinction between notional and real assent "is Newman's most original contribution to the epistemology of belief" ([Price 2013](), p. 316). However important and interesting this distinction is, it should not make us forget that Newman's most striking contribution to the understanding of conviction is that he conceived certainty as what may be called a 'threshold concept'—one of those concepts that are not applicable until they are totally applicable. We may thus say, following Unger's terminology, that, for Newman, assent is an *absolute term*, one that indicates an absolute limit ([Unger 1975](), p. 49).

The problem for gestalt models that make use of absolute terms is that the states of 'seeing as', 'understanding', and 'certitude' *come*, as it was above expressed, *of themselves*, namely, that critical thresholds are connected, however much they may be justified retrospectively, with leaps.

It is, therefore, as if Newman's thesis of the non-entailment of certitude would require a proper explanation, and as if the more plausible candidate to explain belief, absent inefabilist transitions, is an *unconstrained* act of the will.

The problem is how to do justice to the relation between certitude and will without falling into the logical paradigm or arbitrariness. Additionally, we must consider the normative problem of the role played by the *personal* dimension in the epistemology of belief. Might Newman's position lead to a form of proto-constructivism? After all, according to epistemic constructivism, representations are *radically contingent*, meaning that one is always free to represent an object differently from the way the world compels us to perceive it. In this approach, representations at bottom are not rationally structured—they spring from acts which, not being answerable to reasons, are at random and arbitrary. This is the problem of *normative constraint* in believing and what its sources are.

We should begin our exploration by noting that Newman appeals to an "active recognition of propositions as true" (GA, p. 271) to provide an account of the relation between belief and will. What does the *active* nature of an 'active recognition' consist in? How can it help to make a willing culmination of certitude while it is not something that one decides *to do*?

To begin with, it is important to notice that, in Newman's model, reasoning and certitude should not be approached as though they are two distinct factors, and thus, as though 'active recognition' is a way to refer, explaining nothing, to a detached 'happening' that comes after deliberation, cancelling the latter. This is why I prefer to speak of certainty as a *culmination* internal to the process of reasoning. It is to the *entire process of coming to certitude* that we must pay attention.

I have to confess, however, that the previous consideration leaves everything unexplained. It is of no avail to point out that certitude is internal to the entire process if 'recognition' and 'belief' are merely two words for the same phenomenon. The gap, internal or not, remains unexplained. It is only the intriguing 'active' that qualifies the recognition of patterns that keeps me searching for an intelligible connection between inference and assent. What does Newman mean by this little, but crucial word?

It is clear that the primary meaning of what Newman calls an 'active recognition' is that the recognition of patterns cannot be forced upon us. Instead, it is by its nature fully first-personal. But that should not make us forget that Newman's emphasis is on the simplistic and illegitimate opposition of libertarian freedom and the compulsion of logical deduction. His point is that it is the *individual* who reasons and becomes certain, with all his antecedent assumptions. It is hence this emphasis on the individual who "stands by himself" (GA, p. 316) and who, when reasoning, "is his own centre" (GA, p. 271), that makes it explicit that, for Newman, the individual is the *common factor*, or the *driving force*, of every act of reasoning leading to certainty. It is not only the case that belief cannot be against one's will. It is also that the individual remains active during the whole process,

as the element in which, borrowing from Wittgenstein, "arguments have their life" (OC, § 105). Taken in this sense, one might properly say that, for Newman, the argument, to lead to genuine conviction, cannot proceed *without the will*.

But how are we to understand this claim? Let us attempt to shed light on it in terms of the abovementioned simplistic opposition of freedom and compulsion that Newman denies, so as to make room for a *third way* that highlights the alternative opposition of impersonal compulsion and what for Newman is the personal appropriation of reasons. Alternatively, the exploration might be made in terms of common varieties of 'cannot'.

Let us pay attention to a conclusive argument. I can refuse to think about the Pythagorean Theorem, but I am no longer free to think what I will when I attend to it. It is just that I cannot deny its truth. This sharply contrasts with judgments that are guided by personal preference as the result of a choice based on individual needs, desires, and interests. In those cases, the 'cannot' of 'I cannot think of it' is elliptic of 'cannot afford to'. It is apparent that things that one cannot afford to believe are beliefs that can be inhibited by a whole series of different causes (social pressure, intellectual loyalty, intellectual inertia, moral commitment, wishful thinking, etc.). However, the common denominator for all those causes is that they can be overcome (however hard) and their influence resisted. This is why 'I cannot afford to believe it' is usually followed by 'and I will not', the latter expressing a free resolution. However, it makes no sense to say in a *logical* sense that one cannot afford to believe the Pythagorean Theorem.

Epistemic constraint, however, operates beyond the restricted domain of mathematical proofs and conclusive arguments. At the moment the agent attends to all the reasons available to her for p and against q, she is epistemically bound to prefer p over the alternative. It is apparent that even if the agent cannot afford to believe p, she can refuse to think about it, can turn her attention to reasons favouring q while averting her attention from reasons for p, can gradually cultivate her feeling for q, and can even declare that p is not true; however, supposing that at the moment of judging she sees the reasons then involved, she cannot make a judgmental denial of p. Judgment involves no free choice. When one is bound to prefer a proposition as likely to be true, one recognizes it as something which, provided the same conditions, ought to be rationally believed. However, one also recognizes that one cannot (intellectually) help believing it, or, alternatively, that one cannot help preferring it as likely to be true.

According to the previous dialectic, Newman seems to be between the two horns of a dilemma: either belief results from *personal preference*, and it is hence lacking normative constraint (person relativity), *or* the determination of the mind by reasons is as fundamentally heteronomous as its determination by any object (what may be called *the mechanical conception of reasoning*). Newman, however, is anxious to distinguish *constraint* from *compulsion*, and so to argue for a model of reasoning that does justice to the facts that (a) experienced certitude has the character of something that one cannot help believing; (b) this constraint is a *personal constraint*, and hence it is not incompatible with freedom, taken in one way.

The crux of the matter is that Newman's model is based on an *interpretative analogy*, according to which the process of reasoning is like confronting a cryptograph, certainty is like the deciphering of it, and the correctness of this certitude is confirmed by the continuity and the connections that appear everywhere (GA, pp. 255–56).

The point is that the reasoner hits upon the meaning of the letters by which they form intelligible sentences. No doubt remains possible as to the correctness of the deciphering, since the agreement and the consistency cannot be accidental. Similarly, certitude must be confirmed from itself. It must throw light upon the reasons in which it is anchored and bring a heap of discontinuous, non-linear reasons into agreement. Newman therefore

marks the difference between knowing something as a contingent fact like all others, and understanding something as meaningful. Agents are brought to certainty by means of connections of meaningful correspondence.

Notice that, as happens with 'gestalt shifts' of a single, intuitive nature, reaching certitude is something sudden and unexpected, but also something to which one surrenders oneself. This parallels, in a certain sense, logical compulsion.

But this should not mislead us to think that Newman's *hermeneutical paradigm* is reducible to the logical model. In formal reasoning, the reasoner can be gradually guided in the direction of the conclusion, so that he reaches it by following a beaten path, while in informal reasoning, it is the individual who has to guide himself through the jungle of implicit credences, received opinions, intimate experiences, and rational assessments, of which his life consists, towards insight and conviction. More significantly, while in the logical model, the reasoner is conceived of as an abstract intellect, fully detached from the whole of his person, to whom evidence is provided, Newman stresses that the individual, with all his antecedent assumptions, cannot be discounted from the process leading to assent and that he is entirely at one with himself, both in his reasoning and in his certitude, and in complete harmony with those judgments forced upon him in the course of his life.

For Newman, the process does not take place, like volitionalists hint, in a vacuum, nor does it run its full course through a formal, idealized, impersonal medium—rather, it runs its course in *the life of this one person*. The life of each person is, however, a mirror of all lives. This is why conviction can only be something that does not depend upon any particular need or upon any particular preference. The legitimacy of judgments of certainty is relative, not only to a given community, but also to the life of the entire human community to which the person who is certain relates in the hope of finding an echo, a hope shadowed by the fear of misunderstanding. Crucially, Newman's turn to the individual is the turn to the world. Certainty is encountered in, and as directing, our dealings with the world. The discontinuity proper of certitude thus emerges out of experience of the world.

Additionally, that certainty is confirmed from itself means that it is its own rule (GA, pp. 272–73), and thus, that it clearly has a *prescriptive*, rather than merely descriptive, character. Certainty is not based on a relation to external ends and goals. It is truth-centered, and, as such, it constrains the reasoner. But it does not compel him heteronomously, by the force of evidence, whether ordinary or extraordinary. Certitude thus falls in between the view of belief as coerced and the view of belief as accidental.

Here are the essentials of the issues that concern us:

First, if the experience of certitude is not compelled, it is because Newman contends that the informal reasoning that precedes it is interspersed with human life, and thus, that reasons are integrally related to the whole person. That reasons are registered entails for Newman that the impact of life upon us leaves a permanent mark in our gradual development. Reasoning, to lead to conviction, must take place in human life. This explains continuity, and also why Newman's reasons might be distinguished from evidence. Additionally, it makes sense of why he called the process 'free'. It also parallels Wittgenstein's remarks about how "life can educate you to 'believing in God'" (CV, p. 97).

I repeat, certainty is only elicited when a worldview *makes contact with* one's life. Belief is thus essentially interwoven with the vicissitudes of life—some of them are merely practical, while others are more substantial and intimate: self-disgust, yearning for ultimate meaning, the feeling of existential purposeless, physical suffering, and the horizon of death. Crucially, certainty is a function of all those experiences, and what Newman calls 'the personal appropriation' of reasons is nothing else than how each one of us is *embedded* in one's own life. Certainty may have no sense within an epistemological grammar. However, it makes sense in the context of life and action.

Secondly, the problem of the logical model is that it pictures reasoning as a process that only touches our intellectual surface, thus falling short of generating certitude. In this model, conclusive arguments are heteronomous—a remark that agrees with Newman's idea that logical compulsion is felt as 'coming from without'.

My point is that, failing to involve the whole individual, those formal arguments are perceived as notional ways of apprehending truths—as arguments that compel one's assent without one having experienced the fullness of conviction within oneself. Here, one does not believe against one's will. But since conclusions do not penetrate into one's life, one certainly believes them *without one's will*, as not fully at one with oneself. There is a stark contrast between an internal elucidation that will bring about the *dawning* of a meaning aspect not previously appreciated (OC, § 141) and conclusions based on evidence. Additionally, there are philosophical arguments that, although perfectly sound, contradict those convictions that one has adopted in one's everyday life as constitutive of human life. In those cases, the philosopher cannot be convinced and certain of his own philosophy.

Thirdly, the previous consideration makes it clear that Newman, together with providing a serious account of conviction as something that one can only experience at first hand and that is a state of harmony with oneself, is contrasting knowledge with certainty. Certainty refers to patterns and systems of coordinates that are expressed by means of apparent propositions that are not among the data of the *living* cryptograph, and thus, that are "like the axis around which a body rotates" (OC, § 152). They shed, as it were, a glaring light upon the whole. They are intrinsically related, not to knowledge-apt propositions, but rather to worldviews and holistic domains. Again, certainties for Newman are in no sense objects of knowledge.

Finally, it would be tempting to claim that when Newman described the 'directionality' and the 'convergence' of those reasons in which experienced certitude is anchored, he was suggesting that those aspects were immanent to the process of reasoning, as if, like evidence, informal reasons contain (implicitly) the conclusion and the task of the reasoner is merely to follow a given track. It is in this context that a reminder of *discontinuity* is useful. Convergence and directionality can only be appraised after the shift. Otherwise, there is no appreciable distinction between the hermeneutical and the logical models of inference. Relatedly, a broadened concept of willing is also suggested by Newman's view that informal reasoning is inherently motivated by the aim to think what cannot be *epistemically* thought—the disproportionality of inference and assent. Certainty and paradox are thus bound together.

## 4. Back to Wittgenstein: Incommensurability, Hinges, and Religious Commitment

It is a widespread assumption among hinge epistemologists that there is an important *disanalogy* between Wittgensteinian certainties and religious commitments.

On the one hand, it is widely recognized that religious fundamental convictions are not isolated propositions, and so, that if the believer were to lose his faith, then a whole conception of the world would collapse in its entirety. It is in this sense, as a function of their centrality for the believer, that religious beliefs could be legitimately described as hinges (Schönbaumsfeld 2023, p. 34). Additionally, the way in which the religious believer quickly exhausts his amount of supporting evidential reasons is strikingly similar to the manner in which one would reach the bedrock in giving empirical grounds for Moore propositions, or in trying to persuade the sceptic to give assent to hinges.

However, religious fundamental commitments diverge from Wittgensteinian hinges in that the former, unlike the latter, are optional. This has been expressed by arguing, as Pritchard does, that while hinge commitments, because they are necessary for any cognitive

project, are universal, religious commitments are particular ([Pritchard 2022](#), p. 118). This has also been expressed by calling attention to the fact that losing one's faith would not result, unlike what happens with universal hinges, in an annihilation of all criteria.

The crucial factor is that religious belief is not epistemologically fundamental, and thus, that to argue the legitimacy of a religious hinge commitment on the basis of its similarity to Wittgensteinian hinges one would have to argue that the religious conviction has the same universality and necessity that hinges do. It is clear that this eliminates most varieties of the 'parity argument' as advanced by reformed epistemology.[13] It is thus widely accepted that some criteria (and hinges function as rules and criteria) are constitutive of epistemic rationality itself, while others are context-dependent.

I am not entirely convinced that the above assumption is correct, at least if the foregoing claim becomes a stark division, a black-and-white contrast between quotidian and non-quotidian hinges. I also think that attempting to shed light upon Wittgensteinian certainties from Newman's gestaltic model would at least be a useful intellectual exercise, an exercise that might help us to reduce the extent of the noted divergence between religious hinges and structural hinges. Let us remember that, at any rate, Newman's understanding of religious belief was dependent of his understanding of general certainties.

Let us start with the main features of the position that Pritchard maintains, as he has explored this problem in detail.

The first question to face is how religious convictions relate to hinges. On this issue, Pritchard contrasts, as it was noted earlier, the universal character of what he calls 'quotidian hinges' with the context-dependent nature of 'non-quotidian' hinges, such as religious frameworks ([Pritchard 2022](#), p. 120). The term 'universal' is here used in two (related) senses: quotidian hinges are universal in the sense of bracketing the domain of human common sense, epistemic rationality, and the human form of life. They are also universal in the sense of being non-optional. In the first sense, the picture is one in which particular domains are attached, as special grammatical systems, to the main body of human grammar. In the latter sense, quotidian hinges are described as 'visceral' and 'primitive' ([Pritchard 2022](#), p. 93), thus being related to what may be called a 'universal and foundational instinct'. It is clear that, absent a 'religious instinct', religious commitments fail both criteria.

However, a common conception of rationality opens the way to the possibility of contradiction between the quotidian and the non-quotidian systems of reference, and to the consequent necessity of the latter's evaluation. This is the problem of *incommensurability*. There must be some constraint to prevent religious commitments from being arbitrary and fully autonomous. The problem is that, if fully autonomous, context-dependent systems could violate universal rules. Not only 'could' they violate those universal criteria, in fact, and given the paradoxical nature of many fundamental religious beliefs, it is not unusual that this happens.[14] It is not only that religious beliefs do not conform to the manner we are commonsensically guided in thinking and acting. It is rather that, as Wittgenstein notes, referring to St Paul, such beliefs are considered "folly" (LC, p. 58).

At first glance, Pritchard does not seem to be as aware as he should of the above possibility. However, in fairness to Pritchard, it should be noted that he finds a solution to the problem of incommensurability by proposing that it is plausible to think that, while religious commitments reflect core *values* "as opposed to factual commitments" ([Pritchard 2022](#), p. 127), quotidian hinges reflect factual commitments. If this is so, then commitments in non-quotidian contexts can never violate, nor contradict, the factual commitments that hinges codify. Those two kinds of commitments stand, as it were, on a different plane.

Of late, Pritchard has come to stress and fix the foregoing suggestion by hinting at the possibility of extending hinges to a broader set of propositions that capture *axiological commitments* ([Moyal-Sharrock and Pritchard 2024](#), p. 64). This could be seen as a way of

contributing to the agenda of hinge epistemology by helping to provide a comprehensive taxonomy of the varieties of hinges.

The problem of this approach is that it implies a variety of what Cottingham calls the "fruit juicer" model of religious belief (Cottingham 2009, p. 209). This model abstracts the attitudinal from the propositional, thus attempting to capture the essence of religious commitment through a process of distillment that disregards *content* as inessential. However, in respect to religious belief, the *how* and the *what* are integrally related. The crucial point is that religious commitments provide a synoptic world-outlook, a way of rendering the facts of experience comprehensible and a way of adjusting ourselves to them. It is clear that what faith recommends is a practical re-orientation rather than the reception of information, but this change in one's way of life is closely connected with a new way of 'seeing' the world and of appreciating its meaning that is not reducible to a moral mode of life. The believer's convictions guide religious acting along certain channels. The believer treats his beliefs as truths to live by. Religious hinges reflect core values, indeed. But they also reflect an understanding of the world as meaningful. This is why, for Newman, the point of departure of apologetics is always the ordinary life, instead of rational arguments that lead to epistemic claims.

Thus, the problem of incommensurability remains unsolved, and with it a further potential problem emerges: the question of how to distinguish faith from prejudice.

At this point of the discussion, it may be useful to consider that the problem of incommensurability for religious belief might be rephrased in two related ways. On the one hand, it is the question of whether there are criteria that are not generated by religious frameworks that could be legitimately applied to religious commitments so as to ascribe some kind of objectivity to them. The point is that, for religious commitment to be legitimate, immunity to criticism cannot be a necessary feature of it. Otherwise, defenders of religious beliefs may be (rightly) charged with attempting to legitimize uncritical dogmatism. The task here is far from simple. The problem is how to do justice to the fact that, being a form of unconditional commitment, religious commitment is highly intolerant to adverse evidence, while some sort of positive test should be also provided. Relatedly, candidates to become a transcontextual standard cannot be imposed on religious grammar as external, coercive criteria. In other words, the question is how religious commitment can be unconditional, and yet open to a sort of criticism that is available both to insiders and outsiders.

A second way of raising the problem is by means of attending to the way of bringing unbelievers to religious conviction. A passage from CV makes it clear that traditional proofs based on 'evidence' are not sufficient to convert the atheist, and that what must be rather addressed is the latter's personal experience and situation:

> Instruction in a religious faith, therefore, would have to take the form of a portrayal, a description, of that system of reference, while at the same time being an appeal to conscience. (. . .) It would be as though someone were on the one hand to let me see my hopeless situation, on the other depict the rescue-anchor, until of my own accord, or at any rate not led by the hand of the *instructor*, I were to rush up & seize it. (CV, p. 73)

Besides the clear similitudes between Wittgenstein's description of the way to faith and the Socratic method, notice that Wittgenstein is appealing here to conscience, something that parallels Newman's *informal* proof of God from conscience (Hugues 2009, pp. 189–220). It is not only that Newman and Wittgenstein coincide in claiming that conversion is not substantiated by means of formal arguments, but rather that they conceive of the way to faith as resulting from the recognition of what lies at the heart of one's moral experience as its meaning. It is also that this line of reasoning leads them to face the same kind of

objection, which is that even an appeal to intimate experience can be dialectically ineffective insofar as appreciation of its meaning and recognition of its significance are blocked.

Since religious data are only apprehended in the light of the presupposition of God's existence, a religious understanding of experience of the world cannot be taken for granted. We may thus say, following Wittgenstein, that "experience does not direct us to derive anything from experience" (OC, § 130), and so, that the process of conversion seems to be undermined by the very fact that one cannot come to see what it is (allegedly) involved in moral experience without an antecedent understanding of it. Must one be a believer before one can become a believer? Can I only uncover the meaning embedded in my moral experience if I already accept that it is embedded there? At the very least, there seems to be no royal road into faith. Just as, against the backdrop of this criticism, there seems to be no way in which one might entertain an appreciative but cautious stand towards religious belief—a stand that does not involve a positivist denigration of faith, without yet fully adopting it.

In respect to the foregoing problems, the first thing to note is that there is something wrong (and deeply un-Wittgensteinian) about the way hinge epistemologists frame them. As was noted earlier, they provide an account that incorporates *universal* commitments which function as logical rules that allow us to determine whether a belief or a doubt is nonsensical. From their view, not all kinds of hinge acceptances admit the possibility of disagreement. As Pritchard stresses, there are propositions that, in normal circumstances, cannot be genuinely doubted (Pritchard 2022, p. 120). Unlike what happens with religious commitments, a doubt that one has hands would call into question our whole system of reference. It is hence a doubt that cannot be accommodated within the rational worldview.

The crucial point is that on this view human grammar and religious grammar are clearly *distinct* domains. On the one hand, hinge epistemologists appeal to universal standards to guarantee commensurability. But on the other, the universal import of hinge commitments is epistemically insulated from the religious language game, whose autonomy as a non-quotidian domain is recognized. Universal commitments are thus incorporated without achieving commensurability between grammars and domains. Those commitments do not include constraints on their non-quotidian application, so that hinge epistemologists appear to pay lip service to hinge acceptances as a negative test for religious assent.

In other words, the mistake lies in a blindness of sorts to the fact that religious language is *continuous* with human grammar, just as moral, deeply human experience is also continuous with perceptual experience. In that case, the very idea of a quotidian grammar that is purely epistemic and rational, and of hinge commitments exclusive of it, is suspect. As was hinted at earlier, the upshot of all this is that it is only possible to reach transcontextual criteria that are available to religious insiders if one attends to how religious belief is firmly connected with human life, instead of being mesmerized by abstract conceptions of objectivity and universality. The standard must be internally recognized as such, and not imposed from 'alien', self-sustained games.

The fact that Pritchard fails to see this is connected to his assumption that there must be universal hinges. The admission of context dependency for religious belief is what allies Pritchard with classical fideists, inspired by Wittgenstein, like Phillips. Pritchard's corrective contribution to this view lies in the universal notion of hinges. But Pritchard offers no way of showing that those common, invulnerable rules do not collapse into context-dependent determination.

It is also worth emphasizing that Pritchard has recently come to contrast genuine hinge commitments like the paradigm case of one's having two hands (Pritchard 2022, p. 120) with examples of everyday certainties that, even if they appear to be hinges, are logically distinct from them. Those fake hinges are, according to Pritchard, examples of what John

Greco calls "common knowledge" (Greco 2021, pp. 103–25). Crucially, these are cases in which a mistake is perfectly possible, meaning that these propositions point to constant conjunctions that may break down without plunging the system into chaos, so that, to give an example, water over the gas may freeze instead of boil (OC, § 613). A mistake "can be fitted into" the rest of what we know (OC, § 74). But if, in normal circumstances, one were to doubt of having hands, we could not fit this into the rest of what we know. Pritchard substantiates this more nuanced contribution to the taxonomy of hinge acceptances by quoting OC, § 613.[15] What Wittgenstein appears to be saying in this passage is that there is a contrast between cases where failure can be accommodated into our worldview and commitments with the *logical* status of being incompatible with doubt.

However, we need to be wary of supposing, as Pritchard does, that an isolated passage where Wittgenstein made copious use of question marks, qualifications, and tentative language is a sufficient basis on which to build a theory of genuine hinges as opposed to (anachronistic) 'common knowledge'. A common misconception about *On Certainty* consists in the idea that it is a polished text in which notes, instead of recording how things struck Wittgenstein at the moment of thinking them, capture positive doctrines. It is the danger of endowing particular comments with an appearance of generality.

But Pritchard makes a further mistake: he assumes that when Wittgenstein claims that "(i)n certain circumstances a man cannot be making a mistake" (OC, § 155), he is really claiming that genuine hinges are commitments *logically* exempt from doubt. The 'normal' that, according to Pritchard's account, qualifies those circumstances in which a doubt is inconceivable, has the implicit role of downplaying the epistemic significance of the context. It is doubtful, however, that the incompatibility of doubt is a function of certain propositions, as if circumstances were gratuitous adornments. Rather, the incompatibility of doubt is a function of the circumstances, which always make a difference. My point is that it could be plausibly argued that Wittgenstein, instead of targeting hinge commitments, was registering contexts in which no one familiar with the circumstances could think of the possibility of a mistake. If so, Pritchard's confusion results from "a one-sided diet of examples". In any case, universal hinges are far from being undisputable. Small wonder that we end up concluding that the distinction between quotidian and non-quotidian hinges is at best a problematic conception, at worst, an incoherent notion.

What is more, something interesting is going on in Wittgenstein's passage, quoted above, on instruction in faith. Here, again, Wittgenstein is not saying that a demonstrative proof and a mechanical procedure are the means to bring about a situation where the instructed sees things differently, but rather that it is a non-demonstrative form of persuasion that leads him to the principle that rightly interprets the riddle and converts chaos into order. We find, again, a gestaltic and hermeneutical model of persuasion that parallels Newman's method. My suggestion is that it makes sense to extend, in a certain way, this model to non-religious hinges, and so, to argue that, taken in one way, quotidian and religious hinges are on the same plane.

First, what impresses me about Pritchard's picture is that the principle of order (what he calls the "overarching hinge commitment") is manifested in "one's hinge commitments to particular everyday certainties" (Pritchard 2022, p. 99), like having two hands. In this view, the common *Weltbild* is directly captured by a set of particular certainties. We already saw that no clear example can be provided of a context-independent particular certainty that is *intrinsically* related to the system of reference. More significantly, there is also a problem with the stance that Pritchard takes to make this point. It is as if we are able to station ourselves at the limits of the world and capture some determinate truths that make those limits manifest. Of course, those 'truths' are "removed from" ordinary "traffic" (OC, § 210), and yet they can be grasped (and even expressed) within a philosophical context.

Thus, the epistemological context functions here as a vehicle of *indirect communication*, as a way of *showing* determinate contents that cannot be meaningfully expressed in the normal course of life. What this shows is the tendency of hinge epistemologists (i) to collapse the overall worldview into incontestable truths, and (ii) to conceive Wittgenstein's project in OC as that of providing an indirect saying of a determinate 'it'. This flatly ignores that for Wittgenstein it is a *paradox* that experience should seem to be embedded in an ordered whole.

There is a well-known passage in LC, p. 59, in which, regarding the grammar of 'God', Wittgenstein says that, if he were asked whether he understands this word, he would say 'Yes, and no". In speaking this way, Wittgenstein is pointing to the fact that religious language is irreducibly *paradoxical*, meaning that whenever one thinks that one has hit upon a determinate 'it' that is part of the meaning of 'God', one has in fact only hit upon an analogy that does not apply entirely to God, and so, that must be finally revoked. There is hence no detached 'it' to grasp and to communicate.

I take it that the same goes for the grammar of hinge terms, and that, for Wittgenstein, hinge language is, like religious language, ultimately and radically paradoxical. Wittgenstein is thus seeming to put some epistemological doctrines forth but revokes them. He attempts to say something about hinge commitments, only to make us aware that similarities with common epistemological categories suddenly come to an end. Hinges might be described as 'truths', but with the caveats that they are not like empirical truths and that they function like rules. They may have a content, but this particular content expresses an overarching and highly indeterminate überhinge. They are exempt from doubt, but cases in which they could be false without the annihilation of yardsticks are imaginable. Crucially, it is as if any attempt to capture the worldview in particular terms would fail, and thus as if the epistemological illusion would consist in thinking that one might be able to (in Schopenhauer's terms) *represent* a living picture and that there is something that, despite being unable to be directly expressed, is nevertheless logically grasped. The worldview is thus seen as a crust solidified on the surface of experience—and epistemology is conceived as the artificial reconstruction of human experience.

Rather than trying to provide some sort of neo-Kantian deduction of epistemic categories, Wittgenstein suggests that one cannot understand a given category without living in it—that, as happens with religious concepts, the connections which give hinge terms meaning take place in life. Hinges are solidly organized and profoundly animated by common life. If this is right, then what Wittgenstein writes in CV, p. 58, regarding the grammar of 'God'—that the word "does not show *whom* you mean, but what you mean"—can be also extended to hinges: the way you use a certainty does not show the 'it' you mean, but rather what *you* mean. Hence, the impulse behind the intellectualist is our need to be reflected out of a particular stance, and thus, to come to understand ourselves (our life) by means of what we fail to properly mean. With hinge language, like with religious language, we bounce off fragments of sense and bounce back to ourselves in an elucidatory way.

It is true that, as was noted earlier, Wittgenstein refers to 'a description' of the religious system of coordinates as part of the instruction into faith. But this description only makes sense as long as it is a description of our living situation, one that is fully anchored in real life in order to making sense.

It is also true that, unlike what happens with the implicit learning of so-called quotidian hinges, religious beliefs are usually acquired by direct inculcation, and they are taught via explicit statement. However, this disanalogy does not mean that conversion can be intellectually isolated from the vicissitudes of life. On the contrary, what it shows is that early explicit indoctrination is just a way of making room for complex experiences which, though constitutive of the texture of human life, often arrive late in the gradual

development of the individual. It is into the tissue of these experiences that religious beliefs can make sense, and faith can thus become an education for grown-ups. It is precisely for these reasons that it makes sense to regard the religious and the quotidian frameworks as continuous, and so to regard the danger of a radical existential disorientation as having no privileged moorings in the collapse of the regularities of the external world.[16] In any case, Wittgenstein's critique of all the varieties of transcendentalism is accompanied by a recommendation to stay anchored in the world as will, namely, in the language games we play.

The consequence of all this is that it is only possible to develop an *internal* understanding, instead of an external one, of the living transitions between atheism and faith (or the other way around). The solution to the problem of incommensurability must be found within the domain of human life as it is. The illusion of universal criteria results from a metaphysical conception of both objectivity and the ordinary.

## 5. Concluding Remarks

The way out of the quandary consists in heeding Wittgenstein's warning that it is against the backdrop of the gradual development of human life that transcontextual criteria—criteria which are not situated in the abstract realms of the constitutive and the transcendental—can be found. The crucial point is that the appeal to universality and to mere non-contradiction between systems of reference can be seen as a mark against the practical relevance of non-quotidian grammars. However, religious belief is, for Wittgenstein, intrinsically related to a practical re-orientation. It is, again, by means of going back to the rough soil of human life that transitions and positive tests of critizability might be made visible.

As for the problems of how to avoid the charge of dogmatism for religious assent and of how to provide some constraints to religious grammar, the first thing to say is that it is true that religious pictures appear to be particularly unassailable, in the sense that they lie out of reach of demonstration or refutation. However, against the background of a gestaltic form of persuasion that is non-demonstrative without being non-rational, this is not to say that they are invulnerable to criticism.

The power with which a religious picture holds us captive can be broken, and one can be led to abandon it as a whole and to see things in a new way—a way that introduces another order and makes fuller contact with our living reality. Conversely, the religious picture can be so firm and extensive that its defectibility, even if it is possible, cannot be admitted without an effort of the whole mind. The picture's credentials come thus from how it leads to an intimate understanding of an assemblage of data.

The point is that our appreciation of certain things is so unproblematic and so firmly established across a variety of living contexts that one model/picture/rule that ignores or disturbs it is ultimately unacceptable. Our judgments are anchored in a commonality of needs and experiences which are the substantial and transcontextual criteria for the evaluation of worldviews. They are the standards of judgement that outsiders and insiders share and which leave room for defeasibility and for the possibility of criticism. Crucially, those phenomena that are not in dispute mainly refer to *facts* that happen in the human soul—a point that Wittgenstein highlighted when marking that "Christianity is not a doctrine", but "a description of something that actually takes place in human life" (CV, p. 32). Again, objectivity, though it must be independent of particular contexts and personal interests, needs not actually be independent of human life, human consciousness, and human judgment as such. This is an important lesson we received from Wittgenstein.[17]

If we apply these thoughts to the second way of expressing the problem of incommensurability, namely to the question that only moral phenomena apprehended in a particular

way will support a religious conclusion, and so, that religious interpretations are only circularly supported by 'religious data', the implications are analogous.

We might meet this objection by noting that, according to Wittgenstein, gestaltic persuasion happens in the context of trying to make all that surrounds what is basic (the worldview) alive. This suggests, again, that there is neither a clear-cut distinction between 'what is basic' and "what lies around it" (OC, § 144), nor an epistemic asymmetry between those two (alleged) 'factors'. It also indicates that religious persuasion is based upon 'side-on', imaginative ways of throwing light on 'collateral facts', and so, that certainty is swallowed down with its circumstances (OC, § 143).

Crucially, the way to faith is not based on 'religious data', but rather on those transcontextual data that are not in dispute—data which, if the overview to which we are invited is persuasive, are bestowed with such a significance, stability, and order that they come to be appreciated in full for the first time. But those are the gifts of poetic, religious and philosophical geniuses—who instruct us through a therapy that involves some kind of free conversion.

I hope that the foregoing suffices to motivate the thought that Newman's gestaltic turn is at least a plausible candidate to explain certainty and to overcome some epistemological delusions. Hopefully, a second advantage of the foregoing discussion is to bring to the surface some overlooked similitudes between Newman's thought and Wittgenstein's method.

To summarize, I have argued that Newman's emphasis on the gestaltic model of reasoning and the role played by the imagination in informal reasoning is a fruitful starting point for an exploration of the convergence between the *Grammar of Assent* and *On Certainty*. Thus, I think that Wittgenstein, like Newman, challenges both the claim that justification must be neutral and the prejudice according to which any form of persuasion that is not demonstrative is irrational or arational. Added to it, both endorse a view of philosophy as radically therapeutic, and consequently the conception of philosophizing as a radically individualistic activity that demands the active participation of the reasoner. Indeed, the claims that freedom is the essence of philosophy and that in reasoning "the individual is supreme (...) and his own judge" (GA, pp. 277–78) could have come as easily from Wittgenstein as they did from Newman. Clearly, this challenges some widely assumed representations of Wittgenstein's last notes.

**Funding:** This research received no external funding.

**Data Availability Statement:** No new data were created or analyzed in this study. Data sharing is not applicable to this article.

**Conflicts of Interest:** The author declares no conflicts of interest.

## Notes

1  Henceforth, 'OC' will stand for *On Certainty* (Wittgenstein 2004).
2  In references to Wittgenstein's *On Certainty* numbered remarks are cited.
3  Letter to Meynell, 17 November 1869.
4  Henceforth, 'GA' will stand for *Essay in Aid of a Grammar of Assent* (Newman 2008).
5  Newman distinguishes between two kinds of commitment—a non-deliberate adherence and a deliberate adherence (GA, pp. 157–61). A conscious and deliberate assent to our first-order assent is termed a *complex* assent which results from investigation (GA, pp. 157, 159), while the feeling of certitude that we have the duty to investigate and certificate retrospectively is what Newman calls *simple* assent. Complex assents are thus reflective acts that are composed of simple assents and characterized by repose and persistence (GA, p. 167). In addition, Newman is emphatic on the claim that reason and reflection only have a *critical* function, meaning that though they can stifle certainty, they cannot create it (GA, p. 168). He describes the experience of being certain as a bell that strikes (GA, pp. 189–90). [I am grateful to an anonymous reviewer for calling attention to the latter image, which is in accord with the gestaltic model proposed in this paper].The foregoing reasons motivate my suggestion that the certainty we feel and simple assent are, for Newman, synonymous. While the further topic of which conditions have to be met

6    for a reflective appropriation of our certainties is of urgent interest in religious epistemology, it also goes well beyond the aims of the present article. I am grateful to an anonymous reviewer for pressing me on this issue.

6    That assent is independent of inference (what I have called elsewhere *the thesis of the independence of conviction*) is intrinsically related to Newman's anti-evidentialism, namely to his rejection of Locke's two doctrines, that conviction has degrees, and that to be rational, the degree of belief ought to be proportional to the strength of evidence for the proposition to which one assents. On Newman's view, religious belief is characterized as entirely different from ordinary belief closely related to evidence, computation of chance, and knowledge. The significance of this claim lies in its reference to a particular *mode* of pro-attitude characterized by persistence, tenacity, and a spontaneous resistance to change—a kind of adherence that differs from a mere intellectual acceptance of propositions, and so, that suits well the phenomenology of religious conviction. This view is shared by Kierkegaard, Newman and Wittgenstein. I am grateful to an anonymous reviewer for pressing me on this issue.

7    This is the position that was ultimately taken by H. H. Price (2013, pp. 130–56). In my view, Price was wrong in thinking that the paradox formulated by Newman is the second-order paradox that has recently come to fix the agenda of Hinge Epistemology, rather than a first-order paradox that comes from the fact that assent is unconditional, whereas inference (whatever its kind may be) is conditional.

8    This is the common factor shared from therapeutic to theoretical readings of *On Certainty*, as it is argued in (Coliva 2016, pp. 10–12).

9    Henceforth, 'LC' will stand for *Lectures and Conversations on Aesthetics, Psychology, and Religious Belief* (Wittgenstein 2007).

10    Pritchard's conception of the illative sense as an intellectual power is made clear by the following passage: "Newman's idea is that a properly refined illative sense can generate good judgments, and thereby rational beliefs, even when evidential support doesn't meet the Lockean standards. (. . .) In this way, one's illative sense, properly employed, may lead to one trusting a belief where the original evidence is no longer recalled, or sticking with a belief in the face of apparently persuasive counterarguments. (. . .) Of course, one might have a raciocinative faculty that is degraded (. . .) For Newman, then, our basic religious certainties are to be understood such that they could well amount to knowledge, provided they are properly formed via the illative sense. Indeed, they can also count as evidentially grounded, at least in a broad sense of evidence" (Pritchard forthcoming).

11    Newman also expresses this point by describing the illative sense in terms of 'noticing aspects', as follows: "[The illative sense] is a power of looking at things in some particular aspect, and of determining their internal and external relations thereby" (GA, pp. 265–66). Crucially, as it happens with paradigm examples of gestalt-shifts, the illative sense may lead to a "sudden revelation", so that "light breaks in upon us" (GA, p. 266). It is interesting to note that the illative sense is then related to appraisal of meaning, rather than to evidence, inference and knowledge. As Newman makes clear, the logic of language must be supplemented by the more elastic "logic of thought" (GA, p. 281).

12    Henceforth, 'CV' will stand for *Culture and Value* (Wittgenstein 2006).

13    As for a classical reference of parity arguments, see Alvin Plantinga's program in his article "Is Belief in God Rational?" (Plantinga 1979, pp. 7–27).

14    For example, notice how Wittgenstein contrasts quotidian hinges like 'every human being has two human parents' with how Catholics believe that Jesus only had a human mother (OC, §§ 239, 240).

15    Wittgenstein writes in this passage: "If I now say "I know that the water in the kettle on the gas-flame will not freeze but boil", I seem to be as justified in this "I know" as I am in *any*. 'If I know anything I know *this*'. —Or do I know with still *greater* certainty that the person opposite me is my old friend so-and-so? And how does that compare with the proposition that I am seeing with *two* eyes and shall see them if I look in the glass?—I don't know confidently what I am to answer here. —But still there is a difference between the cases. If the water over the gas freezes, of course I shall be as astonished as can be, but I shall assume some factor I don't know of, and perhaps leave the matter to physicists to judge. But what could make me doubt whether this person here is N. N., whom I have known for years? Here a doubt would seem to drag everything with it and plunge it into chaos".

16    That the scientific image of the world might lead to something close to a chaotic disruption of one's yardsticks is illustrated by Darwin's 'horrid doubt', as it was expressed in his letter to William Graham of 3 July 1881: "You would not probably expect anyone fully to agree with you on so many abstruse subjects; and there are some points in your book which I cannot digest. The chief one is that the existence of so-called natural laws implies purpose. I cannot see this. (. . .) Nevertheless you have expressed my inward conviction, (. . .), that the Universe is not the result of chance. But then with me the horrid doubt always arises whether the convictions of man's mind, which has been developed from the mind of the lower animals, are of any value or at all trustworthy. Would any one trust in the convictions of a monkey's mind, if there are any convictions in such a mind?" (Darwin Correspondence Project 2024, Letter no. 13230).It is my view that 'horrid doubts' are more closely connected with images of mechanical processes that undermine our self-conception as self-winding persons and with a deficiency of meaning, than with irregularities of nature.

17    Curiously, the claim that Newman did not require, for objectivity, a necessary correspondence with absolute truth-values independent of human consciousness, has been forcibly argued by Ferreira (1980, pp. 62–70). Here, again, we meet a striking similitude between Newman and Wittgenstein.

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
