# Peer review of "Newman and Wittgenstein on the Will to Believe: Quasi-Fideism and the Ground of Religious Certainty"

_religions, doi:10.3390/rel16020174_

Round 1
Reviewer 1 Report
Comments and Suggestions for Authors
This is a brilliant analysis of Pritchard and a very well documented engagment with Wittgenstein and indeed Newmam. However, the discussion of Newman suggests that some terms are synonymous (certainty, assent etc) and doesnt really demonstrate this enough. Also Newmans suggestion that assent is independent of inference (even if argued to be an isolated statement) raises further questions . All that being said, this is a really important contribution to tge debate.
Author Response
Dear reviewer,
Sincere thanks for carefully reading my paper and for your thoughtful comments and suggestions for improving it.
Reviewer 1 has been so kind as to praise the paper as "a brilliant analysis of Pritchard and a very well documented engagement with Wittgenstein and indeed Newman", as well as a "really important contribution" to the debate. I am deeply grateful to the reviewer for those appreciative comments, as I am grateful to the reviewer for the recommendations to improve the paper.
Reviewer 1 has recommended a more extended explanation of why I consider that certainty and assent are, for Newman, synonymous. I have added footnote 5 (in the attached file) to elaborate this point.
Reviewer 1 has also recommended that the claim that assent is independent of inference should be better explained. I have provided a better explanation in footnote 6 (in the attached file).
I am deeply grateful to the reviewer for pressing me on those issues. (Please, find the revisions in red).
Reviewer 2 Report
Comments and Suggestions for Authors
I consider this a brilliant paper. The gestaltic interpretation of Newman's take on certainty presented here, constitutes a breath of fresh air in a series of misunderstandings that have pestered Newman studies. The author illuminatingly describes the various issues and questions raised by (common interpretations of) Newman's understanding of certainty, and very helpfully presents a different model. For me, the extent to which he identifies the commonly raised issues as problems of (the lens of) epistemology, rather than of Newman's project as such, is key. Also, I regard the critique of Pritchard's take on hinge epistemology as apt and welcome.
The paper combines the clear exposition of a complex (to this reader) argument with genuine depth of observation - a rare quality in philosophic papers these days.
The paper shows extensive familiarity with the work of the authors dealt with - especially Newman and Wittgenstein, but also James. Neither Wittgenstein nor Newman are quoted extensively, but to the reader familiar with their ideas, this is a boon rather than a drawback. Therefore, I do not recommend more citations, and would advise against them if other reviewers thought they were necessary. The same goes for discussion of the recent literature. Where Newman is concerned, one could think of recent work by Fred Aquino and Logan Gage on Newman's purported fallibilism etc. But again, I don't think this would add much to an already substantial article.
In re Newman, I have a few suggestions for further consideration or expansion of the article, that I would like to share, but that need not be incorporated into the article in order to be publishable.
1. The author might want to think about the distinction Newman draws between certainty as a quality of propositions and certitude as a mental state (GA 344). What, if any, bearing does this distinction have on the argument of this paper? I wonder especially how the author would explain certainty as a quality of propositions.
2. On fallibilist interpretations of Newman (like that of Aquino & Gage), Newman's concept of the indefectibility of certitude is hard to make sense of and left largely unexplained. On the gestaltic model proposed in this article, it could be explained much better (once you see the duck, you cannot unsee it - sure, you can see the rabbit again, but you can always shift your attention and see the duck again). This might be worth pursuing.
3. Newman describes the experience of being certain as a bell that strikes (GA 233, 236): "The sense of certitude may be called the bell of the intellect". I think this image can be fruitfully applied to the gestaltic model proposed in this paper, and might be worth adding.
Now as to the few very minor issues I would actually like to see addressed. (1)) Perhaps too much is expected of the reader in terms of familiarity with the technicalities of present-day epistemology, such as the repeated use of the term "k-apt propositions" or the unexplained use of abbreviations for Wittgenstein's works. (2) On occasion, there are slight linguistic mistakes or things that are slightly unclear. I list the ones that struck me below.
l94: 'the experience of a proposition' - unclear what this means
l98-99: "simple certainty we feel when believing which I mean" - unclear construction
l130-132: unnecessarily complex formulation
l157: "which is" should read "which it is"
l203: "portray" should be "portrayal"
l282-283: "neatly distinction"
l319: "would be no" should be "there would be no"
l378: replace "but" with "by"
l381-383: unnecessarily complex (double negation + illegitimate)
l489-490: revise grammar
l508: remove "it"
l706: "struck" for "stroke"
l722: preposition missing before "Wittgenstein"
Author Response
Dear reviewer,
Sincere thanks for carefully reading my paper and for your thoughtful comments and suggestions for improving it.
Reviewer 2 has been so kind as to praise highly the paper, claiming that it is "a brilliant paper" and "a breath of fresh air in a series of misunderstandings" regarding Newman. I am deeply grateful to the reviewer for the encouraging words, which touch my heart. I am also deeply grateful for the suggestions to improve the paper.
Some of those suggestions "need not be incorporated into the article". I have incorporated the image of certainty as the bell of the intellect in footnote 5. However, I have opted (following the optional nature of those comments) not to incorporate a further discussion on certainty as a quality of propositions, nor to deal with fallibilist interpretations of Newman. It is not only that those topics are so vast as to require a second paper (or a series of papers). It is also that I am currently working on a paper that deals with those problems from the standpoint of complex, reflective assent, and of forms of doubt that may be compatible with conviction.
As for the suggestions that should be addressed, I have substituted 'knowledge apt propositions' for 'k-apt propositions', as well as I have explained (on page 2, in red) what I mean by 'knowledge apt beliefs'. I have explained the use of abbreviations for Wittgenstein's works. I have also corrected all the typos, linguistic mistakes, and grammatical infelicities that the reviewer has been so kind as to list.
I am deeply grateful to the reviewer for pressing me on those issues. (Please, find attached the revised manuscript with revisions in red).